# Determinants of modern contraceptive use among married and sexually active unmarried women aged 15–49 years old in Cambodia: How are geographical and socio-demographic factors associated with access?

Samnang Um[1]*, Pall Chamroen[1,2], Chantrea Sieng[3], Sovandara Heng[4], Grace Marie Ku[5,6]

1 The National Institute of Public Health, Phnom Penh, Cambodia, 2 KHANA Center for Population Health Research, Phnom Penh, Cambodia, 3 Outpatient Department, Calmette Hospital, Phnom Penh, Cambodia, 4 Department of Mental Health and Substance Abuse, Khmer Soviet Friendship Hospital, Phnom Penh, Cambodia, 5 Department of Public Health, Institute of Tropical Medicine, Antwerp, Belgium, 6 Faculty of Medicine & Pharmacy, Vrije Universiteit Brussel, Brussels, Belgium

* umsamnang56@gmail.com

## Abstract

Using modern contraceptives is essential for promoting economics, limiting the space of pregnancies, and improving maternal and newborn health outcomes. Despite progress in Cambodia, disparities in contraceptive use persist, particularly across socio-demographic and geographic groups. This study examined the prevalence and determinants of modern contraceptive use among married and sexually active women of reproductive age (15–49 years) in Cambodia, focusing on geographic and socio-demographic factors. This cross-sectional analysis of individual women's data from the 2021–2022 Cambodia Demographic and Health Survey (CDHS), including those of 13,492 women. Modern contraceptive use was defined as the self-reported current use of methods such as sterilization, IUDs, implants, pills, injectables, condoms, and emergency contraception. Descriptive and logistic regression models were used to examine associations with factors such as geographic accessibility (distance to healthcare, transportation, residence, regions), socio-demographic characteristics, and healthcare access. The prevalence of modern contraceptive use was 45%, higher in rural areas (47%) than in urban areas (41.4%). Women living 20–40 minutes from healthcare facilities were less likely to use contraceptives (AOR = 0.87, 95% CI: 0.75–0.99) compared to those within 20 minutes. Motorcycle ownership was positively associated with contraceptive use (AOR = 1.29, 95% CI: 1.09–1.53). Higher odds of use were observed among women with two children (AOR = 29.33, 95% CI: 20.04–42.91) or three or more children (AOR = 30.76, 95% CI: 20.50–46.16). Contraceptive use was lower among wealthier women and those aged 35–49 compared to younger women (15–19). These findings highlight the need for targeted interventions,

**Data availability statement:** The Cambodia Demographic and Health Survey 2021-2022 datasets are publicly available from the website: (URL: https://www.dhsprogram.com/data/dataset_admin).

**Funding:** The authors received no specific funding for this work.

**Competing interests:** The authors have declared that no competing interests exist.

**Abbreviations:** AOR, Adjusted odds ratio; ANC, Antenatal care; BMI, Body mass index; CDHS, Cambodia Demographic Health Survey; CI, confidence intervals; EA, Enumeration areas; GIS, Geographic Information System; PPS, Probability proportional to size; WHO, World Health Organization; RGC, Royal Government of Cambodia; VIF, Variance inflation factor

including expanded rural healthcare, transportation solutions, and tailored awareness campaigns, to improve equitable access to family planning services and support Cambodia's health goals.

## Introduction

Modern contraceptive methods are significant predictors of preventing maternal morbidity and mortality and improving newborn survival rates. These methods also help in reducing adolescent pregnancies and unsafe abortions, and may contribute to economic growth. Most condoms are effective in preventing HIV and other sexually transmitted infections [1]. However, according to a 2022 report of the United Nations Population Fund (UNFPA), despite the availability of modern contraceptives, an estimated 257 million women who wish to avoid pregnancy are not using safe, modern methods of contraception [2]. There are approximately 121 million unintended pregnancies globally each year [2].

Cambodia has historically experienced a high fertility rate coupled with high maternal, newborn, infant, and under-five mortality rates [3]. However, Cambodia's health indicators have notably improved over the past 20 years. Data from the 2021–22 Cambodia Demographic and Health Surveys (CDHS) show that the total fertility rate declined from 3.8 births per woman in 2000 to 2.7 births per woman in the 2014 survey and remained stable at 2.7 in the 2021–22 survey. Neonatal mortality has declined from 37 deaths per 1,000 live births in 2000–8 deaths per 1,0000 live births in 2021–22. Infant mortality has declined from 95 deaths per 1,000 live births in 2000–12 deaths per 1,000 live births, while the under-five mortality rate (U5M) has decreased from 124 deaths per 1,000 live births in 2000–16 deaths per 1,000 live births in 2022. Additionally, maternal mortality has declined from 488 to 154 per 100,000 live births between 2000 and 2021–22 [4]. This achievement can be attributed to the country's concerted efforts to increase women's access to maternal health services, particularly the initiative to increase the use of modern contraceptives from 33% in 2014 to 45% in 2021–22. Institutional births have markedly increased, from 19.3% to 98%, while the proportion of pregnant women attending four or more antenatal care (ANC) appointments increased considerably, from 9% to 86.1%, between 2000 and 2021–22 [4]. Nevertheless, the country's maternal mortality still needs to be reduced by more than half by 2030, to achieve the SDG 3.1 global target of less than 70 per 100,000 live births by 2030 [5]. The preceding, and including achieving newborn mortality of less than 12 per 10,00 live births and under-five mortality of less than 25 per 10,00 live births, are among the high-priority aims of the Royal Government of Cambodia (RGC) in ending preventable deaths. Contraceptive use significantly reduces maternal mortality by decreasing the number of pregnancies a woman experiences, thus reducing her exposure to pregnancy-related risks [6–8]. It also prevents high-risk, high-parity births, which are linked to complications such as hemorrhage and obstructed labor. By enabling women to space and limit pregnancies, contraceptive use lowers the likelihood of these risks and maternal deaths [6–8].

Additionally, it allows women to plan pregnancies, ensuring better timing and preparation, which further reduces health complications [6–8]. Ultimately, family planning is a critical tool in reducing maternal mortality by preventing high-risk pregnancies and minimizing overall exposure to pregnancy-related dangers [6–8].

Previous studies in Cambodia on contraceptive use using CDHS 2021–22 indicated that 12% of currently married women and 60% of sexually active but unmarried women have an unmet need for family planning, while 42% of current married women who are not using any method of contraception and report wanting to delay the next child or limit childbearing, which varies depending on geographical area and socio-economic status [4]. The prevalence of married women reporting using modern contraceptives is at 45% in 2021–22, with similarly low reported prevalence of 47% in rural areas and remote areas [4]. In other Asian countries, earlier studies have demonstrated higher prevalence of modern contraceptive use at 57.2% in the Philippines in 2017 and 55.7% in Myanmar in 2016 [9]. A previous literature review pooling DHS from South and Southeast Asia indicated that women with motorcycles were more likely to report modern contraceptive use, with an adjusted odds ratio (AOR = 1.60, 95% CI 1.54-1.66) compared with women who did not own a motorcycle [10]. Another study showed that women who were exposed to media were more likely to use contraception in the Philippines (AOR = 2.24, 95% CI = 1.42-3.54) and Myanmar (AOR 1.39, 95% CI = 1.15-1.67) [9]. A study investigating the relationship between distance to nearby facilities and modern contraceptive use among married women in rural Ethiopia found a significant decline in utilization rates as the distance to service delivery points increased. According to the analysis using geo-referenced data, 41.2% of women living less than 2 kilometers from a facility used modern contraceptives, compared to 27.5% at 2 to 3.9 kilometers, 22.0% at 4 to 5.9 kilometers, and 22.6% for those residing 6 kilometers or more away (p-value < 0.01) [11]. In Cambodia, the population-based household study using data from CDHS 2014 showed that women aged 35 years or older were less likely to use modern contraceptives use than those aged 15–34 years old (AOR = 0.73, 95% CI = 0.61-0.87) [12]. Similarly, lower odds of modern contraceptive use were found among women who completed secondary or higher education compared with those who had no education (AOR = 0.70, 95% CI = 0.54- 0.90) and women living in rural areas compared to those living in urban areas (AOR = 0.73, 95% CI = 0.60-0.88) [12,13]. These earlier findings highlight the influence of socio-demographic factors on modern contraceptive use in Cambodia.

To date, challenges associated with access to modern contraceptives, specifically distance, mode of transportation, geographical region, place of residence (urban/rural), health insurance membership, internet use, exposure to media, and socio-economic status, have not yet been explored in Cambodia. While previous analyses of the CDHS 2005–2014 data examined general factors associated with modern contraceptives use [12,14], this study specifically focuses on aforementioned geographical and socio-demographic factors as potential determinants of access influencing use in the most recent 2021–2022 CDHS data. Unlike the descriptive overview provided in the CDHS 2021–2022 report [4], our study employs multiple logistic regression model to disentangle the independent effects of these factors on modern contraceptive use among women of reproductive age (15–49 years of age) in Cambodia. Understanding the geographical underpinnings of low modern contraceptive use can support policymakers to better allocate resources to improve access to family planning services, which is essential for reducing unplanned pregnancies and enhancing maternal and child health outcomes. This study aimed to assess the prevalence of modern contraceptive use among married and sexually active women aged 15–49 in Cambodia and examine its association with specific socio-demographic and geographical factors.

## Methods

### Study design

This cross-sectional study made use of the women's dataset (Individual Recode file) from the Cambodia Demographic and Health Survey (2021–2022), a nationally representative household survey conducted every five years [4]. The 2021–2022 CDHS dataset is the most recent nationally representative data available for Cambodia, providing an up-to-date understanding of contraceptive use patterns and associated factors. Analyzing this recent data allows us

to identify current challenges and inform timely interventions, building upon findings from earlier CDHS analyses by focusing on specific geographical and access-related determinants that have not been extensively explored previously. The survey employed a two-stage stratified cluster sampling method to collect data on women and men aged 15–49, their children, and household characteristics from all provinces. In the first stage, clusters, or enumeration areas (EAs), representing the entire country (urban and rural) are randomly selected from the sampling frame using probability proportional (PPS) to cluster size. In the second stage, a complete listing of households was selected from each cluster using equal probability systematic sampling. Women aged 15–49 living in the selected household who had given birth in the last five years preceding the survey were invited to be interviewed face to face using a standard survey questionnaire collecting a variety of information on health indicators such as maternal health care service utilization, reproductive health services, history of fertility, maternal and child health nutrition. The CDHS employs a rigorous sampling methodology to ensure national representativeness. A total of 21,270 households were selected, with 19,496 women aged 15–49 included in the survey; the response rate was high at 98.2%. The CDHS is designed to yield precise national and sub-national estimates for key health indicators, and the achieved sample size exceeds typical requirements for such analyses. The large sample size of women provides sufficient statistical power to detect associations between the independent variables and modern contraceptive use. The final CDHS 2021–2022 report details have been published [4].

## Study population

A total of 13,492 eligible married and sexually active unmarried women aged 15–49 years were included in the analysis. The remaining 6,004 women were excluded because they were never married, or are widowed or divorced, and are not sexually active in the 30 days preceding the survey, as determined using the CDHS variable v528 (time since last sexual intercourse) [4].

## Outcome variable

The outcome variable was the current use of modern contraceptive methods among married and sexually active unmarried women aged 15–49. The CDHS collected self-reported information on the current use of family planning methods, including modern contraceptives such as male and female sterilization, intrauterine devices (IUDs), injectables, implants, contraceptive pills, male and female condoms, emergency contraception, the standard day's method, and the lactational amenorrhea method. Traditional methods included rhythm or periodic abstinence, withdrawal, folks, and herbs [10,12,15]. However, the CDHS does not specifically track dual-method use; instead, modern and traditional methods are recorded separately. The original variables were categorized as dichotomous: current use of modern methods was coded as **1**, and non-user and use of traditional contraceptive methods were coded as **0**.

## Independent variables

We included data on geographic factors, including distance to health facilities, healthcare utilization and barriers, sociodemographic factors, and insurance coverage.

Geographic accessibility factors consist of place of residence (coded as 1 = urban and 2 = rural), and Cambodia geographic regions were coded as Phnom Penh capital city = 1 (reference), Plains = 2, Tonle Sap = 3, Coastal/sea = 4, and Mountains = 5 [16]. Distance was measured in minutes for respondents to reach the nearest health facility from home, using a motorized vehicle such as a car/truck, a public bus, a motorcycle/scooter, or a boat with a motor and was computed based on the women self-reported estimate of the time taken using their primary mode of transport to the nearest health facility [4]. It was coded as less than 20 minutes = 1 (reference), 20–40 minutes = 2, and 40 and above = 3. Mode of transport to healthcare facilities was coded as car = 1 (reference), motorcycle = 2, bicycle = 3, and walking = 4. Access to a motorcycle was measured based on household ownership coded as (no = 0 yes = 1). Access to healthcare facilities and

family planning visits in the past years is coded as (no = 0, yes = 1). Perceived healthcare barriers are coded as a dichotomous variable with No barriers = 0 and 1 or more barriers [17].

Socio-demographic variables consist of women's age in years (15–19, 20–24, 25–34, 35–49). Women's education attainment was coded as an ordinal variable where 1 = no education (reference), 2 = incomplete primary, 3 = complete primary, 4 = incomplete secondary, 5 = complete secondary, and 6 = higher. Employment status coded as not working = 1 (reference), professional = 2, sales = 3, agricultural = 4, services = 5 and manual labor = 6.

The DHS calculates the households' wealth index using principal component analysis (PCA) variables for household assets (e.g., ownership of television, telephone, refrigerator, bicycle, motor, and car); and dwelling characteristics (e.g., type of flooring, source of water, and sanitation facilities) [4,18]. The resulting weighted scores are then divided into five wealth quintiles, each representing 20% of the households [4,18]. In this study, the household wealth index was categorized according to the original DHS classification: poorest = 1 (reference), poorer = 2, middle = 3, richer = 4, richest = 5 [4,18]. Health insurance coverage, including public and private insurance, is coded as no = 0 and yes = 1 [19].

Media exposure was assessed by asking women about their frequency of reading newspapers/magazines, listening to the radio, and watching television. Responses were categorized into 'None' (no exposure to any of these media), 'One' (exposure to one type of media), and 'Two or more' (exposure to two or more types of media) [20]. Internet use was a binary variable indicating whether the respondent reported using the internet.

## Data analysis

Statistical analysis was performed using STATA SE V18. We applied the DHS standard sampling weight variable (v005/1,000,000). We used the survey-specific STATA command "svy" for descriptive, chi-square, and logistic regression analysis, accounting for the complex survey design. Descriptive statistics were utilized to describe the weighted frequency and percentage distributions. The provincial prevalence of modern contraceptive use was estimated by using ArcGIS software version 10.3 [13]. The Cambodian shapefile of provinces was obtained from the United Nations for Coordination of Humanitarian Affairs (OCHA) at (https://data.humdata.org/dataset/cod-ab-khm). Data quality assurance and minor data cleaning followed standard procedures in preparation for statistical analysis [14].

We used the Chi-square test to explore the association between the independent variables (including geographic, healthcare, and sociodemographic factors) and modern contraceptive use. We also used simple logistic regression to analyze associations between modern contraceptive use and geographic, healthcare, and sociodemographic factors. Results are reported as crude odds ratios (COR) with 95% confidence intervals (CI) and corresponding p-values. Independent variables associated with modern contraceptive use at p-value ≤ 0.10 [21], or identified as potential confounder variable on the literature(for example, women's age and household wealth index) were included in the final multiple logistic regression analyses [12,22]. Results from the final adjusted model are reported as adjusted odds ratios (AOR) with 95% CI and corresponding p-values.

Multicollinearity of the independent variables, distance to healthcare, geographical regions, place of residence, household wealth index, education attainments, occupations, and motorcycle ownership, was evaluated using the variance inflation factor (VIF) for the regression coefficients [23].

To evaluate the potential effect modification of statistically significant associations in the adjusted analysis, these were further visualized as predicted probabilities using STATA 18, employing the **margins** command to estimate and visualize with marginal plot [24].

## Ethical issues

The original study protocol of CHDS 2021–2022 was approved by the Cambodia National Ethics Committee for Health Research on 10 May 2021 (Ref: **083 NECHR**) and the Institutional Review Board (IRB) of ICF in Rockville, Maryland,

USA. More details on the informed consent process during the CDHS data collection can be found in the CDHS 2021–2022 report [4]. Generally, trained enumerators obtained informed consent from all participants before the interviews were conducted. Participants were informed about the study's objectives, procedures, potential risks and benefits, and their right to refuse or withdraw participation at any time. Confidentiality and anonymity were assured. All personal identifiers of study participants have been removed from the CDHS database, which is available upon request through the DHS website at (https://dhsprogram.com/data/available-datasets.cfm).

## Results

### Characteristics of the study population

The characteristics of the study population regarding access to modern contraceptive use, including socio-demographic and geographical factors of the 13,492 women included in the study presented (Table 1). Most women (48.7%) of the married and sexually active women of reproductive age included in this analysis were 35 years or older, 14.6% had four or more children, and 6.9% had no children. Accessibility to healthcare facilities, with 80% of women living within 20 minutes of a facility, and motorcycles were the most common mode of transportation (92.6%). A total of 59.9% reside in rural areas, of whom 13.9% live in mountainous regions. Only 10.9% had completed at least secondary or higher education, and 14.0% reported no formal education. Almost 19% of the women live in the poorest households, 18.1% live in poorer households, and 21.3% of women reported not having an occupation. Notably, only 22% of women had health insurance coverage, while 89.5% reported motorcycle ownership, 60.8% use the internet, and 54.8% reported no exposure to any of the accessed media (newspapers/magazines, radio, television), 43% had visited health facilities in the last 12 months, and 41% reported no perceived barriers to healthcare access.

### Prevalence of modern contraceptive use by provinces

The overall prevalence of modern contraceptive use among currently married and sexually active unmarried women was 45% (95% CI: 43.3-46.1). Meanwhile, 45% (95% CI: 43.6-46.4) of currently married women and 27.6% (95%CI: 19.6,37.4) of sexually active unmarried women were reported to use modern contraceptives. These echo the earlier study's result on modern contraceptive use among currently married women and sexually active unmarried women, analyzing the same dataset [4]. The prevalence of modern contraceptive use varied across provinces, ranging from 25.3% in Pursat to 56.1% in Banteay Meanchey and 57.4% in Ratanak Kiri (Fig 1).

### Chi square analysis of factors associated with modern contraceptive use

Table 2 shows the results of chi-square analyses of geographic, healthcare, and sociodemographic factors as related to modern contraceptive use in Cambodia. Women living more than 40 minutes from a health facility had a higher prevalence of contraceptive use compared to those within 20 minutes and within 20–40 minutes (52.0% vs. 44.4% and 44.1%; p = 0.034). Use increased with age, from 31.6% among women aged 15–19 to 45.3% for those aged 35–49 (p < 0.001). Parity showed a positive association, with contraceptive use rising from 4.6% among childless women to 51.7% for those with two children (p < 0.001). Educational attainment showed an inverse trend, with higher use among women with no education (46.8%) compared to those with higher education (35.8%; p < 0.001). The wealth index followed a similar inverse pattern, decreasing from 49.1% in the poorest to 42.4% in the richest group (p = 0.013). Women in agricultural occupations reported the highest use (50.1%), while those in service jobs had the lowest (30.3%; p < 0.001). Motorcycle ownership was associated with higher use (45.3% vs. 39.6%; p = 0.003). In contrast, internet users reported lower use than non-users (43.6% vs. 46.4%; p = 0.037). Rural women had higher contraceptive use (46.9%) than urban women (41.4%; p < 0.001). Regional differences were significant, with the lowest use in Phnom Penh (36.3%) and the highest in Tonle Sap and Coastal regions (48.3%; p < 0.001).

**Table 1. Characteristics of Individual Married and Sexually active women of reproductive age (15–49 years) in Cambodia (2021-2022) (N = 13,492 weighted).**

| Variables | Freq. | Percent |
|---|---|---|
| **Distance to Healthcare*** | | |
| < 20 minutes | 10,652 | 79.0 |
| 20-40 minutes | 2,276 | 16.9 |
| > 40 minutes | 565 | 4.2 |
| **Transport to Healthcare** | | |
| Motorcycle | 12,488 | 92.6 |
| Walking | 649 | 4.8 |
| Car | 153 | 1.1 |
| Bicycle | 120 | 0.9 |
| Boat | 81 | 0.6 |
| **Age in years** | | |
| 15-19 | 335 | 2.5 |
| 20-24 | 1,384 | 10.3 |
| 25-34 | 5,205 | 38.6 |
| 35-49 | 6,568 | 48.7 |
| **Parity** | | |
| None | 935 | 6.9 |
| One | 2,864 | 21.2 |
| Two | 4,777 | 35.4 |
| Three | 2,944 | 21.8 |
| Four or more | 1,972 | 14.6 |
| **Educational attainment** | | |
| No Education | 1,893 | 14.0 |
| Incomplete Primary | 4,687 | 34.7 |
| Complete Primary | 1,393 | 10.3 |
| Incomplete Secondary | 4,053 | 30.0 |
| Complete Secondary | 768 | 5.7 |
| Higher | 698 | 5.2 |
| **Wealth index** | | |
| Poorest | 2,532 | 18.8 |
| Poorer | 2,441 | 18.1 |
| Middle | 2,655 | 19.7 |
| Richer | 2,921 | 21.6 |
| Richest | 2,944 | 21.8 |
| **Occupation** | | |
| None | 2,880 | 21.3 |
| Professional | 757 | 5.6 |
| Sales | 3,327 | 24.7 |
| Agricultural | 2,678 | 19.8 |
| Services | 294 | 2.2 |
| Manual labor | 3,315 | 24.6 |
| **Covered by health insurance** | | |
| No | 10,522 | 78.0 |
| Yes | 2,970 | 22.0 |

*(Continued)*

**Table 1.** (Continued)

| Variables | Freq. | Percent |
|---|---|---|
| **Motorcycle Ownership** | | |
| No | 1,423 | 10.5 |
| Yes | 12,070 | 89.5 |
| **Internet Use** | | |
| No | 5,285 | 39.2 |
| Yes | 8,207 | 60.8 |
| **Media exposure** | | |
| None | 7,393 | 54.8 |
| One | 4,278 | 31.7 |
| Two or more | 1,821 | 13.5 |
| **Visited health facility last 12 months** | | |
| No | 7,690 | 57.0 |
| Yes | 5,803 | 43.0 |
| **Healthcare barriers** | | |
| No barrier | 5,528 | 41.0 |
| 1+barrier | 7,964 | 59.0 |
| **Place of residence** | | |
| Urban | 5,404 | 40.1 |
| Rural | 8,088 | 59.9 |
| **Geographical region*** | | |
| Phnom Penh | 1,994 | 14.8 |
| Plain | 4,607 | 34.1 |
| Tonle Sap | 4,175 | 30.9 |
| Coastal | 845 | 6.3 |
| Plateau/Mountain | 1,871 | 13.9 |

**Notes:** Survey weights are applied to obtain weighted percentages. **\*Distance to Healthcare** in minutes using a motorized vehicle such as a car/truck, a public bus, a motorcycle/scooter, or a boat with a motor. **\*Phnom Penh capital city**; **\*Plains**: Kampong Cham, Tbong Khmum, Kandal, Prey Veng, Svay Rieng, and Takeo; **\*Tonle Sap**: Banteay Meanchey, Kampong Chhnang, Kampong Thom, Pursat, Siem Reap, Battambang, Pailin, and Otdar Meanchey; **\*Coastal/sea**: Kampot, Kep, Preah Sihanouk, and Koh Kong; **\*Mountains:** Kampong Speu, Kratie, Preah Vihear, Stung Treng, Mondul Kiri, and Ratanak Kiri.

### Determinants of modern contraceptive use in multiple logistic regression

No significant multicollinearity was identified, as the highest correlation coefficient was less than 0.6, and the Variance Inflation Factors (VIF) values ranged from 1.04 to 1.65. The goodness-of-fit of the final model was evaluated using the Hosmer-Lemeshow test, which indicated an acceptable fit (p=0.5751). Additionally, the model's discriminative ability was assessed through the Area Under the Curve (AUC), which was 0.6390 (Fig 2), suggesting a weak but discernible ability to distinguish between users and non-users of modern contraception.

In the final multiple logistic regression model, 11 covariates were kept. These covariates were: distance to healthcare, women's age, parity, education, wealth index, occupation, motorcycle ownership, internet use, healthcare barriers, place of residence, and geographical region.

Table 3 shows the results of the adjusted odds ratios (AOR) of the independent factors associated with modern contraceptive use among married and sexually active women of reproductive age. The following variables were noted to be significant:

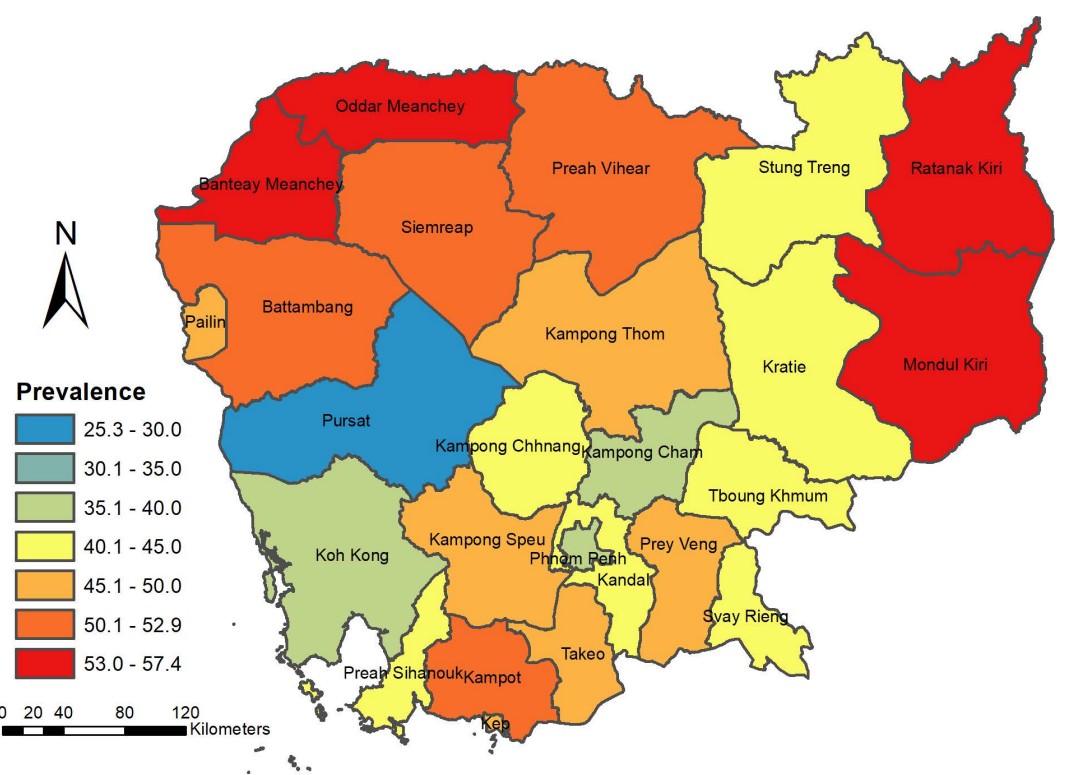

**Fig 1.  Prevalence of modern contraceptive use among Married and Sexually active unmarried women of reproductive age (15–49 years) in Cambodia (2021-2022).**

Women living 20–40 minutes away from healthcare facilities have significantly lower odds of using modern contraception (AOR = 0.87, 95% CI: 0.75–0.99). Conversely, women living more than 40 minutes from healthcare facilities show slightly higher but statistically non-significant odds of modern contraceptive use (AOR = 1.10, 95% CI: 0.86–1.42; p-value = 0.435). Women aged 20–24 years have significantly lower odds of using modern contraception compared to adolescents aged 15–19 (AOR = 0.73, 95% CI: 0.52–1.02; p-value = 0.069). Although the preceding is only marginally significant, we noted that women aged 35–49 years have significantly lower odds of using contraception (AOR = 0.41, 95% CI: 0.29–0.57) as compared to adolescents aged 15–19 years. Additionally, women aged 25–34 (AOR = 0.54, 95% CI: 0.39–0.74) and 35–49 (AOR = 0.41, 95% CI: 0.29–0.57) have significantly lower odds of contraception use as compared to women aged 15–19. Parity was found to be the strongest predictor of contraceptive use. Women with two children (AOR = 29.33, 95% CI: 20.04–42.91), three children (AOR = 30.76, 95% CI: 20.50–46.16), and four or more children (AOR = 27.77, 95% CI: 18.47 - 41.75) have significantly higher odds of using modern contraception compared to women without children. This highlights the strong influence of having children on the adoption of modern contraception. Occupation is also a significant factor influencing contraceptive use. Women in manual labor occupations have significantly higher odds of using contraception (AOR = 1.69, 95% CI: 1.46–1.97). Similarly, women in agricultural work show significantly higher odds (AOR = 1.50, 95% CI: 1.31–1.73). Women in sales and professional occupations have slightly higher odds of using contraception (AOR = 1.23, 95% CI: 1.07–1.41). Motorcycle ownership is significantly associated with higher odds of contraceptive use (AOR = 1.29, 95% CI: 1.09–1.53). Women living in the Tonle Sap region have significantly higher odds of using contraception compared to those from Phnom Penh (AOR = 1.74, 95% CI: 1.23–2.46). Educational attainment and wealth index were not significantly associated with modern contraceptive use in the final model.

PLOS Global Public Health

**Table 2. Chi-square analysis of geographic factors, healthcare utilization, sociodemographic factors, and insurance coverage with modern contraceptive use in Cambodia (2021-2022), (N=13,492 weighted).**

| Variables | Not Using Modern Contraception (N=7,462) | | Using Modern Contraception (N=6,031) | | P-value |
|---|---|---|---|---|---|
| | Freq. | % | Freq. | % | |
| **Distance to Healthcare** | | | | | |
| < 20 minutes | 5,918 | 55.6 | 4,734 | 44.4 | 0.034 |
| 20-40 minutes | 1,273 | 55.9 | 1,003 | 44.1 | |
| > 40 minutes | 270 | 47.8 | 294 | 52.0 | |
| **Transport to Healthcare** | | | | | |
| Motorcycle | 6,869 | 55.0 | 5,619 | 45.0 | |
| Walking | 392 | 60.4 | 257 | 39.6 | |
| Car | 99 | 64.7 | 54 | 35.3 | 0.230 |
| Bicycle | 64 | 53.3 | 56 | 46.7 | |
| Boat | 38 | 46.9 | 44 | 54.3 | |
| **Age in years** | | | | | |
| 15-19 | 229 | 68.4 | 106 | 31.6 | <0.001 |
| 20-24 | 835 | 60.3 | 549 | 39.7 | |
| 25-34 | 2805 | 53.9 | 2400 | 46.1 | |
| 35-49 | 3,592 | 54.7 | 2,976 | 45.3 | |
| **Parity** | | | | | |
| None | 892 | 95.4 | 43 | 4.6 | <0.001 |
| One | 1,788 | 62.4 | 1,076 | 37.6 | |
| Two | 2,307 | 48.3 | 2,470 | 51.7 | |
| Three | 1,448 | 49.2 | 1,496 | 50.8 | |
| Four or more | 1,027 | 52.1 | 945 | 47.9 | |
| **Educational attainment** | | | | | |
| No Education | 1,008 | 53.2 | 885 | 46.8 | <0.001 |
| Incomplete Primary | 2,505 | 53.4 | 2,182 | 46.6 | |
| Complete Primary | 756 | 54.3 | 637 | 45.7 | |
| Incomplete Secondary | 2,269 | 56.0 | 1,784 | 44.0 | |
| Complete Secondary | 476 | 62.0 | 291 | 37.9 | |
| Higher | 448 | 64.2 | 250 | 35.8 | |
| **Wealth index** | | | | | |
| Poorest | 1,289 | 50.9 | 1,242 | 49.1 | 0.013 |
| Poorer | 1,324 | 54.2 | 1,117 | 45.8 | |
| Middle | 1,496 | 56.3 | 1,158 | 43.6 | |
| Richer | 1,656 | 56.7 | 1,265 | 43.3 | |
| Richest | 1,696 | 57.6 | 1,248 | 42.4 | |
| **Occupation** | | | | | |
| None | 1,751 | 60.8 | 1,129 | 39.2 | <0.001 |
| Professional | 444 | 58.7 | 313 | 41.3 | |
| Sales | 1,880 | 56.5 | 1,447 | 43.5 | |
| Agricultural | 1,336 | 49.9 | 1,343 | 50.1 | |
| Services | 206 | 70.1 | 89 | 30.3 | |
| Manual labor | 1,715 | 51.7 | 1,600 | 48.3 | |
| **Covered by health insurance** | | | | | |
| No | 5,837 | 55.5 | 4,685 | 44.5 | 0.583 |
| Yes | 1,625 | 54.7 | 1,346 | 45.3 | |

*(Continued)*

**Table 2.** (Continued)

| Variables | Not Using Modern Contraception (N=7,462) | | Using Modern Contraception (N=6,031) | | P-value |
|---|---|---|---|---|---|
| | Freq. | % | Freq. | % | |
| **Motorcycle Ownership** | | | | | |
| No | 858 | 60.3 | 564 | 39.6 | 0.003 |
| Yes | 6,603 | 54.7 | 5,467 | 45.3 | |
| **Internet Use** | | | | | |
| No | 2,832 | 53.6 | 2,454 | 46.4 | 0.037 |
| Yes | 4,630 | 56.4 | 3,577 | 43.6 | |
| **Media exposure** | | | | | |
| None | 4,098 | 55.4 | 3,295 | 44.6 | 0.150 |
| One | 2,309 | 54.0 | 1,969 | 46.0 | |
| Two or more | 1,055 | 57.9 | 766 | 42.1 | |
| **Visited health facility** | | | | | |
| No | 4,276 | 55.6 | 3,414 | 44.4 | 0.579 |
| Yes | 3,186 | 54.9 | 2,617 | 45.1 | |
| **Healthcare barriers** | | | | | |
| No barrier | 3,155 | 57.1 | 2,374 | 42.9 | 0.026 |
| 1+barrier | 4,307 | 54.1 | 3,657 | 45.9 | |
| **Place of residence** | | | | | |
| Urban | 3,165 | 58.6 | 2,239 | 41.4 | <0.001 |
| Rural | 4,297 | 53.1 | 3,792 | 46.9 | |
| **Geographical region** | | | | | |
| Phnom Penh | 1,271 | 63.7 | 723 | 36.3 | <0.001 |
| Plain | 2,604 | 56.5 | 2,002 | 43.5 | |
| Tonle Sap | 2,159 | 51.7 | 2,016 | 48.3 | |
| Coastal | 437 | 51.7 | 408 | 48.3 | |
| Plateau/Mountain | 991 | 53.0 | 881 | 47.1 | |

**Notes:** Survey weights are applied to obtain weighted percentages.

Fig 3 illustrates the predicted probabilities of modern contraception use based on motorcycle ownership, place of residence, and parity derived from the multiple logistic regression model (Table 3). These factors were chosen because they directly influence access to healthcare and family planning decisions. Motorcycle ownership enhances mobility, particularly in rural areas where healthcare services are less accessible [10]. Place of residence affects healthcare availability, with rural women often facing significant challenges in accessing contraceptives [10]. The number of living children reflects reproductive intentions, as women with more children are more likely to seek contraception [22]. The findings indicate that motorcycle ownership increases contraceptive use in both urban and rural areas, with a strong effect in rural settings. Additionally, women with three or more children have a higher probability of using contraception, regardless of residence. These results highlight the significance of transportation, geographic accessibility, and family size in shaping family planning access, particularly in rural areas. Furthermore, three-way interactions were explored involving occupation, geographical regions, and parity. However, these analyses revealed that the interaction terms were not statistically significant, indicating that occupation and parity were the primary factors associated with modern contraceptive use, irrespective of the region in Cambodia.

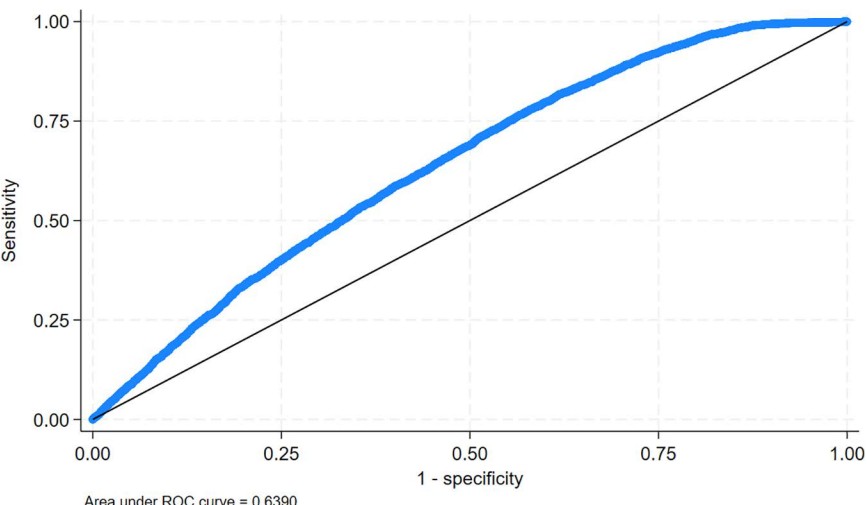

Area under ROC curve = 0.6390

**Fig 2. ROC Curve and Area Under the Curve (AUC) for Modern Contraceptive Use Following Multiple Logistic Regression Analysis.**

## Discussion

This study examined the geographic factors that may affect access to healthcare facilities (specifically distance to healthcare, transportation, residence, regions) and socio-demographic characteristics that may influence modern contraceptive use among married and sexually active women of reproductive age in Cambodia. The overall prevalence of modern contraceptive use was 45%. This is lower than the reported global contraceptive prevalence of modern methods estimated at 58.7% for married or sexually active women [25].

Contextualizing Cambodia's contraceptive trends within the broader Southeast Asian region reveals both similarities and differences. While the overall prevalence of modern contraceptive use in Cambodia (45%) is lower than in some neighboring countries like the Philippines (57.2% in 2017) and Myanmar (55.7% in 2016) [9,10], the influence of factors like and socio-economic status on contraceptive use has been observed across the region. For instance, the positive role of motorcycle ownership in facilitating access aligns with findings from other South and Southeast Asian countries [10]. Understanding these regional patterns can help Cambodia learn from successful interventions implemented elsewhere and tailor its strategies accordingly.

Further, rural areas reported higher usage (47%) than urban areas (41.4%). Regional disparities within Cambodia were also prominent, with higher usage in the Tonle Sap and Coastal regions (predominantly rural) and lower prevalence in Phnom Penh (36.3%; urban area). The findings regarding the lower use among urban women differ from the observations from the 2014 CDHS data, suggesting an emerging pattern that needs to be addressed [12]. This can be attributed to interventions targeting rural populations in Cambodia, as has been done in some low- and middle-income countries and achieving notable uptake [9,26]. Our findings of lower use among urban women underscore the importance of addressing regional inequities in family planning services, particularly in urban areas where differing cultural expectations, perceptions of need, or healthcare quality may drive lower usage.

Geographic accessibility was a significant factor. Women living within 20 minutes of a healthcare facility were less likely to use contraceptives than those residing more than 40 minutes away, who had slightly higher odds of use (AOR = 1.10). This finding is counterintuitive to the evidence from Ethiopia, where greater distances corresponded to lower use but where higher motivation to access services were noted among those more committed to family planning [27]. In Cambodia, this phenomenon could reflect self-selection among highly motivated women to travel longer distances for care. Women living

**Table 3. Multiple logistic analysis of factors associated with modern contraceptive use in Cambodia.**

| Variables | Unadjusted (N = 13,492 weighted) | | (N = 13,251, weighted) | |
| --- | --- | --- | --- | --- |
| | OR | 95% CI | AOR | 95% CI |
| **Distance to Healthcare** | | | | |
| < 20 minutes | Ref. | | Ref. | |
| 20-40 minutes | 0.98 | (0.87 - 1.12) | 0.87** | (0.75 - 0.99) |
| > 40 minutes | 1.36** | (1.07 - 1.73) | 1.10 | (0.86 - 1.42) |
| **Transport to Healthcare** | | | | |
| Car | Ref. | | | |
| Motorcycle | 1.49 | (0.81 - 2.75) | | |
| Bicycle | 1.60 | (0.73 - 3.52) | | |
| Boat | 2.12** | (1.05 - 4.26) | | |
| Walking | 1.2 | (0.57 - 2.52) | | |
| **Age in years** | | | | |
| 15-19 | Ref. | | Ref. | |
| 20-24 | 1.42** | (1.03 - 1.96) | 0.73* | (0.52 - 1.02) |
| 25-34 | 1.85*** | (1.38 - 2.47) | 0.54*** | (0.39 - 0.74) |
| 35-49 | 1.79*** | (1.35 - 2.37) | 0.41*** | (0.29 - 0.57) |
| **Parity** | | | | |
| None | Ref. | | Ref. | |
| One | 12.40*** | (8.51 - 18.08) | 13.95*** | (9.55 - 20.39) |
| Two | 22.07*** | (15.31 - 31.80) | 29.33*** | (20.04 - 42.91) |
| Three | 21.30*** | (14.59 - 31.09) | 30.76*** | (20.50 - 46.16) |
| Four or more | 18.95*** | (12.97 - 27.69) | 27.77*** | (18.47 - 41.75) |
| **Educational** | | | | |
| No Education | Ref. | | Ref. | |
| Incomplete Primary | 0.99 | (0.86 - 1.14) | 1.04 | (0.91 - 1.20) |
| Complete Primary | 0.96 | (0.80 - 1.15) | 1.04 | (0.86 - 1.26) |
| Incomplete Secondary | 0.89 | (0.77 - 1.05) | 1.02 | (0.86 - 1.22) |
| Complete Secondary | 0.70*** | (0.53 - 0.92) | 0.88 | (0.66 - 1.17) |
| Higher | 0.64*** | (0.49 - 0.83) | 0.85 | (0.60 - 1.22) |
| **Wealth index** | | | | |
| Poorest | Ref. | | Ref. | |
| Poorer | 0.88* | (0.76 - 1.00) | 0.93 | (0.80 - 1.07) |
| Middle | 0.80*** | (0.71 - 0.91) | 0.91 | (0.79 - 1.06) |
| Richer | 0.79*** | (0.68 - 0.92) | 0.98 | (0.83 - 1.17) |
| Richest | 0.76*** | (0.64 - 0.92) | 1.17 | (0.95 - 1.44) |
| **Occupation** | | | | |
| None | Ref. | | Ref. | |
| Professional | 1.09 | (0.89 - 1.35) | 1.56*** | (1.16 - 2.08) |
| Sales | 1.19*** | (1.05 - 1.35) | 1.23*** | (1.07 - 1.41) |
| Agricultural | 1.56*** | (1.37 - 1.78) | 1.50*** | (1.31 - 1.73) |
| Services | 0.67** | (0.45 - 0.99) | 0.83 | (0.53 - 1.31) |
| Manual labor | 1.45*** | (1.25 - 1.67) | 1.69*** | (1.46 - 1.97) |
| **Covered by health insurance** | | | | |
| No | Ref. | | | |
| Yes | 1.03 | (0.92 - 1.15) | | |

*(Continued)*

**Table 3.** (Continued)

| Variables | Unadjusted (N = 13,492 weighted) | | (N = 13,251, weighted) | |
|---|---|---|---|---|
| | OR | 95% CI | AOR | 95% CI |
| **Motorcycle Ownership** | | | | |
| No | Ref. | | Ref. | |
| Yes | 1.26*** | (1.08 - 1.47) | 1.29*** | (1.09 - 1.53) |
| **Internet Use** | | | | |
| No | Ref. | | Ref. | |
| Yes | 0.89** | (0.80 - 0.99) | 0.98 | (0.87 - 1.10) |
| **Media exposure** | | | | |
| None | Ref. | | | |
| One | 1.06 | (0.96 - 1.17) | | |
| Two or more | 0.90 | (0.77 - 1.06) | | |
| **Visited health facility** | | | | |
| No | Ref. | | | |
| Yes | 1.03 | (0.93 - 1.14) | | |
| **Healthcare barriers** | | | | |
| No barrier | Ref. | | Ref. | |
| 1 + barrier | 1.13** | (1.01 - 1.25) | 1.05 | (0.94 - 1.18) |
| **Place of residence** | | | | |
| Urban | Ref. | | Ref. | |
| Rural | 1.25*** | (1.10 - 1.41) | 1.12* | (0.98 - 1.27) |
| **Geographical region** | | | | |
| Phnom Penh | Ref. | | Ref. | |
| Plain | 1.35** | (1.03 - 1.78) | 1.27 | (0.95 - 1.69) |
| Tonle Sap | 1.64*** | (1.25 - 2.16) | 1.54*** | (1.16 - 2.05) |
| Coastal | 1.64*** | (1.23 - 2.19) | 1.54*** | (1.15 - 2.07) |
| Plateau/Mountain | 1.56*** | (1.18 - 2.07) | 1.39** | (1.03 - 1.87) |

**Notes:** Survey weights are applied to obtain weighted percentages.

**\*\*\* p < 0.01, \*\* p < 0.05, \* p < 0.10**

20–40 minutes away showed lower odds of use (AOR = 0.87), suggesting that intermediate distances may pose unique barriers, such as inconsistent transportation options or partial access to healthcare. This suggests a non-relationship between distance and contraceptive use, potentially indicating that while moderate distance poses a barrier, those living very far might have different – or higher – motivations or access pathways. Addressing these barriers through mobile health clinics or expanded service availability in moderately accessible areas could enhance coverage [28].

Transportation played a pivotal role in determining contraceptive use. Women who owned a motorcycle were significantly more likely to use modern contraceptives (AOR = 1.29), highlighting mobility as a crucial enabler of healthcare access in Cambodia. Similar findings have been reported in other Southeast Asian settings, where motorcycle ownership increased access to healthcare and family planning services by facilitating travel to distant facilities [10]. In rural areas, where public transportation is scarce, interventions such as transportation subsidies or community vehicle-sharing programs could radically impact family planning service utilization [29].

Socio-demographic factors, including age, parity, and occupation, significantly influenced contraceptive use. Younger women aged 15–19 reported the highest odds of contraceptive use, with decreasing odds in the older age groups, 25–34

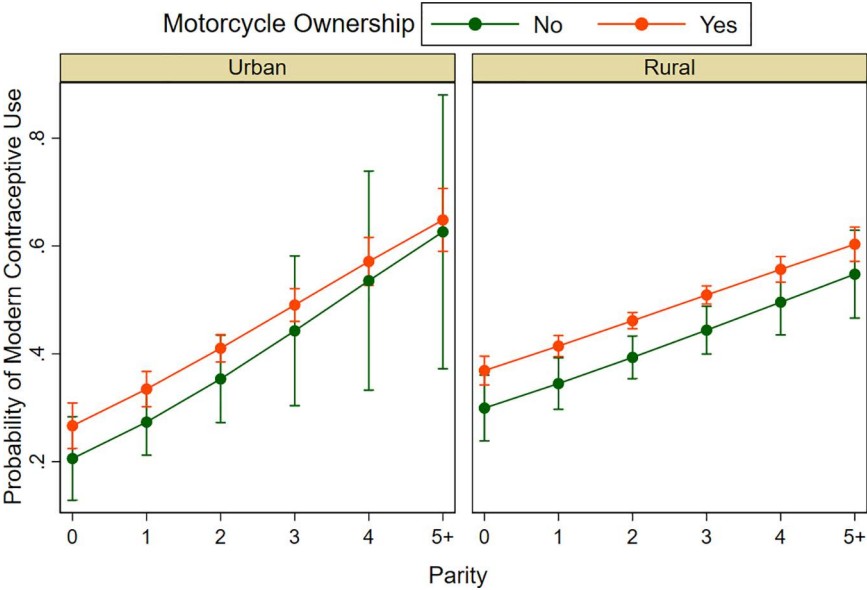

**Fig 3. Predicted probability of modern contraception use.** Illustrates the estimated probabilities based on a three-way interaction: motorcycle ownership, place of residence (rural/urban), and parity.

(AOR = 0.54) and 35–49 (AOR = 0.41). This aligns with regional findings, where targeted interventions for younger women have increased contraceptive uptake to address early pregnancies [14]. The findings regarding the lower use among older women echo observations from the 2014 CDHS data [12], suggesting a persistent pattern that needs to be addressed. This could be attributed to factors such as changing fertility desires, perceived lower risk of pregnancy especially among the 35–49 age group, or potentially a shift towards traditional methods. Parity was the strongest predictor, with women having two (AOR = 29.33) or three or more children (AOR = 30.76) significantly more likely to use contraceptives than childless women. This indicates that family planning efforts are most effective among women with higher parity, which may reflect on family decisions on the number of children they wish to have. Messaging could then be directed towards proper spacing between children and the ideal number of children, to reinforce modern contraceptive use [12].

Contrary to the bivariate analysis, higher educational attainment and wealth index were not significantly associated with modern contraceptive use in the multiple logistic regression models. This suggests that the initial associations observed in the chi-square tests might be confounded by other factors included in the model, such as age, parity, and geographical location.

Economic and occupational factors further shape contraceptive use. Women engaged in manual labor (AOR = 1.69), agricultural work (AOR = 1.50), and sales (AOR = 1.23) were more likely to use contraceptives compared to unemployed women. These findings suggest that economically active women may prioritize smaller families to balance work and household responsibilities. In contrast, the wealth index showed an inverse relationship with lower contraceptive use among wealthier women. Women in the richest households were less likely to use contraceptives compared to those in the poorest households, which may reflect cultural norms or preferences for larger families in affluent groups or having the privilege and the means to take time off work to carry a pregnancy and take care of the offspring. This trend contrasts with findings from Nepal and India, where wealthier women reported higher contraceptive use due to better access and education [12,30], highlighting a unique dynamic in Cambodia.

Finally, while media exposure has been identified as a critical enabler of family planning in other countries [9], its limited impact in Cambodia suggests the need for more culturally tailored messaging strategies. Educational campaigns should

address specific misconceptions and promote the benefits of modern contraceptives, particularly in underrepresented groups such as wealthier and older women.

This study has several implications for policy and practice. The lower modern contraceptive use in urban Phnom Penh compared to other regions warrants specific attention. Policy recommendations for urban areas could include strengthening family planning counseling and services in urban health centers, addressing potential cost barriers or misconceptions that might exist in urban settings, and utilizing urban-specific media channels to promote awareness and access to modern contraceptives [31]. While urban areas like Phnom Penh are generally assumed to have better access to healthcare services, including family planning, the lower contraceptive use in such settings is unexpected. This discrepancy highlights potential underlying barriers—such as misinformation, provider bias, or gaps in service delivery—that remain unaddressed. Further research is needed to identify the specific factors contributing to this lower use in Phnom Penh. Additionally, the findings concerning lower use among older women echo observations from the 2014 CDHS data [12], suggesting a persistent pattern that needs to be addressed. Interventions targeting older women of reproductive age should be put in place.

This study's strengths include using nationally representative data and GIS mapping to illuminate regional disparities. However, this study has several limitations. The cross-sectional nature of the data limits our ability to establish causal relationships between the independent variables and modern contraceptive use. The reliance on self-reported data may be subject to recall bias and social desirability bias. Furthermore, while we controlled for several key socio-demographic and geographical factors, there may be other unmeasured confounders, such as cultural norms, partner preferences, and access to information through informal networks, that could influence contraceptive use.

In conclusion, geographic, socio-demographic, and economic factors significantly influence modern contraceptive use in Cambodia. Addressing disparities through targeted interventions will be essential to achieving equitable access to family planning services and advancing Cambodia's national health goals. Policymakers must prioritize investments in healthcare infrastructure, transportation solutions, and awareness campaigns to ensure no woman is left behind in accessing essential reproductive health services.

## Supporting information

**S1 Data.**
(DTA)

## Acknowledgments

We thank the DHS program for giving permission to use the CDHS 2022 datasets.

## Author contributions

**Conceptualization:** Samnang Um, Chantrea Sieng, Sovandara Heng.

**Data curation:** Chantrea Sieng.

**Formal analysis:** Samnang Um, Pall Chamroen, Grace Marie Ku.

**Investigation:** Samnang Um.

**Methodology:** Samnang Um, Pall Chamroen, Sovandara Heng, Grace Marie Ku.

**Project administration:** Samnang Um, Chantrea Sieng, Sovandara Heng.

**Supervision:** Grace Marie Ku.

**Validation:** Samnang Um.

**Visualization:** Samnang Um.

**Writing – original draft:** Samnang Um, Pall Chamroen, Chantrea Sieng, Sovandara Heng, Grace Marie Ku.

**Writing – review & editing:** Samnang Um, Grace Marie Ku.

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
