## [Decision Letter · Decision Letter 0]

25 Mar 2025

PGPH-D-25-00309

Determinants of Modern Contraceptive Use among Women of reproductive age in Cambodia: How are geographical and socio-demographic factors associated with access?

Dear Dr. Um,

Thank you for submitting your manuscript to PLOS Global Public Health. After careful consideration, we feel that it has merit but does not fully meet PLOS Global Public Health’s publication criteria as it currently stands. Therefore, we invite you to submit a revised version of the manuscript that addresses the points raised during the review process.

Please do respond to the detailed comments from the reviewers.

We look forward to receiving your revised manuscript.

Kind regards,

Anushka Ataullahjan

Guest Editor

Journal Requirements:

1. Figure 1: please (a) provide a direct link to the base layer of the map (i.e., the country or region border shape) and ensure this is also included in the figure legend; and (b) provide a link to the terms of use / license information for the base layer image or shapefile. We cannot publish proprietary or copyrighted maps (e.g. Google Maps, Mapquest) and the terms of use for your map base layer must be compatible with our CC-BY 4.0 license.

Additional Editor Comments (if provided):

Reviewers' comments:

Reviewer's Responses to Questions

**Comments to the Author**

1. Does this manuscript meet PLOS Global Public Health’s publication criteria?

Reviewer #1: Partly

Reviewer #2: Yes

2. Has the statistical analysis been performed appropriately and rigorously?

Reviewer #1: I don't know

Reviewer #2: Yes

3. Have the authors made all data underlying the findings in their manuscript fully available (please refer to the Data Availability Statement at the start of the manuscript PDF file)?

Reviewer #1: Yes

Reviewer #2: Yes

4. Is the manuscript presented in an intelligible fashion and written in standard English?

Reviewer #1: Yes

Reviewer #2: Yes

Reviewer #1: The paper is generally well-written with a good discussion. However, there are several definitions and analysis steps (including the variable selection and model comparison/evaluation) which need to be clarified. The authors should also suggest reasons for the differing relationships inferred from the chi-squared tests, univariate and multivariate logistic regression.

Comments:

• p.4, lines 108-113: These results from a past study using CDHS 2014 data can be mentioned again in the Discussion since they are similar to what is observed in terms of age, rural-urban and education factors for modern contraceptive use in this study.

• p.4, lines 115-119: This is a bit confusing. Is there a distinction made between “access” and “use”? What are considered “general factors”? Are age, education and rural/urban factors included? It would be good to clarify how this study is different from [4] as well.

• p.6, line 167-169: Could you clarify how the distance/time to reach the nearest health facility from home was computed? Were the survey participants first asked how they travel to the nearest health facility and then the distance/time calculated based on their chosen mode of travel?

• p.6, line 179: Define “DHS”. Please elaborate more on the principal components analysis. Did you use the first principal component as a wealth index and used quantiles to define the categories, e.g. poorest to richest?

• p.8, line 234: What kind of media was considered?

• p.12, line 279-281: Is the full set of variables considered in Table 2? How were the variables selected in the final multivariate logistic regression model? Please provide a goodness-of-fit measure and explain how the final model was obtained.

• p.12, line 288-296: Is the distance away from healthcare facilities confounded with the urban/rural variable? Why does the lower odds of women aged 35-49 of using modern contraception compared to adolescents aged 15-19 years conflict with the positive association of age with use in the chi-squared tests and unadjusted logistic regression on p.10? Is age confounded with e.g. parity? It would be important to interpret the adjusted odds ratios by holding all the other variables in the model fixed. On a related note, the insignificance of education and wealth in the multivariate logistic regression suggests that their effects on use could be explained by other covariates in the model.

• Figure 2: How are the probabilities calculated? Did you fix the values of the other covariates in the model? It would be good to describe the interpretation of the figure more. Conditional on parity, it seems that the difference in probability of use for women who own motorcycles or not is not significant in urban areas and only significant in rural areas for women who have 1-3 children.

• p.14, lines 317-318: In the multivariate logistic regression, living in a rural area is associated with higher use of modern contraceptives. This conflicts with the interpretation of the trends in Figure 2 when only 4 variables are examined.

• p.14, lines 320-321: Can you test for the “strong effect” of motorcycle ownership in rural settings in the multivariate logistic regression using an interaction term between the two variables?

• p.16, lines 379-382: Could you check these inferences further by for example, examining the relationship between parity and wealth in the data?

Minor comments:

• p.2, line 39-40 (Abstract, Results): The AORs are missing the 95% CIs.

• p.2, line 53-54: Please add a reference or elaborate on how modern contraceptive methods may contribute to economic growth.

• p.3, line 99: “odd” is missing a “s”.

• p.4, line 130: What does “IR” stand for?

• p.5, line 166: There is a missing “a” for “Coastal”.

• p.9, line 249: There is a missing space between “95%” and “CI”.

• p.12, line 289: The p-value is missing.

• p.15, line 351: Please add the p-value and/or 95% CI to the AORs for geographic accessibility.

Reviewer #2: Formal Reviewer Comment for PLOS Global Public Health

Manuscript ID**: PGPH-D-25-00309

Title: Determinants of Modern Contraceptive Use among Women of Reproductive Age in Cambodia: How Are Geographical and Socio-Demographic Factors Associated with Access?

Reviewer’s Assessment

Summary of the Manuscript

The manuscript presents an important and well-structured study on modern contraceptive use in Cambodia, using nationally representative data from the Cambodia Demographic and Health Survey (CDHS) 2021–2022. The study highlights key socio-demographic and geographic determinants influencing contraceptive use, providing valuable insights for public health interventions. The mixed statistical approach, including descriptive and logistic regression analyses, enhances the study’s rigor.

While the research is well-executed and relevant to global public health discussions, some areas require clarification and refinement to improve transparency, methodological rigor, and the clarity of interpretations. Below, I outline specific recommendations for improvement.

Major Comments

1. Study Design & Justification-

The manuscript does not explicitly state that it is a *cross-sectional* study in the methodology section. Clarifying this is essential to set appropriate expectations regarding causality.

The rationale for using this dataset should be expanded upon. How does it build on or differentiate itself from prior research using CDHS data?

2. Methodological Transparency

Sample Size Justification: It is unclear whether a *power calculation* was conducted to determine the adequacy of the sample size. If not, please explain how the current sample size is sufficient.

Selection Bias: The study focuses only on married and sexually active women, excluding unmarried and inactive women. While this is logical, the paper should explicitly *acknowledge this limitation*.

Confounder Control: More information is needed on how *confounders were selected* for adjustment in the regression models. Was a *multicollinearity check* conducted?

3. Statistical Analysis & Interpretation

The finding that *motorcycle ownership significantly increases contraceptive use*(AOR = 1.29, 95% CI: 1.09–1.53) is important. The authors should emphasize how *mobility constraints* affect healthcare access in rural areas.

The paper should explain *why women living further from healthcare facilities showed higher contraceptive use*, as this contradicts expected trends. This counterintuitive result should be further analyzed could it be due to self-selection bias, where women highly motivated to use contraception travel longer distances?

The study finds that contraceptive use **increases with parity** (AOR for women with three or more children = 30.76). This suggests that **family planning efforts are more effective after women have had multiple children**, but strategies to promote earlier adoption should be discussed.

A *subgroup analysis* (e.g., wealthier vs. poorer women, urban vs. rural) would strengthen the findings by identifying variations in contraceptive access.

4. Discussion & Policy Implications

The *novelty of the study* should be clearly stated. What is new about these findings compared to prior studies?

The *policy implications *need to be more concrete. Instead of general recommendations, can the authors suggest *specific interventions* (e.g., mobile health clinics, targeted subsidy programs)?

The discussion should better integrate findings from *other Southeast Asian countries* to contextualize Cambodia’s contraceptive trends.

Expand the discussion on age-related patterns**:

-Why do older women use contraception less? Is it fertility-related, preference for traditional methods, or reduced health-seeking behavior?

Provide clearer policy recommendations for urban areas**, where contraceptive use is unexpectedly lower.

5. Limitations & Ethical Considerations

The *limitations section* should explicitly mention *self-reported data bias*, potential *causality issues due to the cross-sectional nature*, and *unmeasured confounders* such as cultural norms.

Ethical considerations are well-addressed, but **more details on informed consent** during CDHS data collection would strengthen the manuscript’s transparency.

Minor Comments

The abstract should explicitly state the *study design* (cross-sectional).

Tables should be *formatted for readability*, particularly those with dense numerical data.

There are some *minor grammatical issues* a language edit would improve clarity.

Improve variable categorization explanations* (e.g., why specific cutoff points for distance to healthcare were chosen).

Discuss dual-method contraceptive use (modern + traditional), if relevant.

Recommendation

** Accept with Major Revisions and Minor Revisions** – The study is methodologically sound and contributes to public health knowledge, but revisions are needed to **clarify methodology, enhance result interpretation, and strengthen policy implications**.

I appreciate the authors' effort in conducting this research and look forward to seeing an improved version of the manuscript.

**Do you want your identity to be public for this peer review?** For information about this choice, including consent withdrawal, please see our Privacy Policy

Reviewer #1: No

Reviewer #2: **Yes: ** Betelihem Asrat Biru

---

## [Decision Letter · Decision Letter 1]

11 Aug 2025

Determinants of Modern Contraceptive Use among Married and Sexually Active UnmarriedWomen aged 15–49 years old in Cambodia: How are geographical and socio-demographic factors associated with access?

PGPH-D-25-00309R1

Dear Dr. Um,

We are pleased to inform you that your manuscript 'Determinants of Modern Contraceptive Use among Married and Sexually Active UnmarriedWomen aged 15–49 years old in Cambodia: How are geographical and socio-demographic factors associated with access?' has been provisionally accepted for publication in PLOS Global Public Health.

Please note: you may want to add a space between "unmarried" and "women" in the title for clarity.

Best regards,

Julia Robinson

Executive Editor

Reviewer Comments (if any, and for reference):

Reviewer's Responses to Questions

**Comments to the Author**

Reviewer #1: All comments have been addressed

publication criteria?

Reviewer #1: Yes

3. Has the statistical analysis been performed appropriately and rigorously?

Reviewer #1: Yes

4. Have the authors made all data underlying the findings in their manuscript fully available (please refer to the Data Availability Statement at the start of the manuscript PDF file)?

Reviewer #1: Yes

5. Is the manuscript presented in an intelligible fashion and written in standard English?

Reviewer #1: Yes

Reviewer #1: (No Response)

**Do you want your identity to be public for this peer review?** For information about this choice, including consent withdrawal, please see our Privacy Policy

Reviewer #1: No
